# Attitudes towards Vaccinations in a National Italian Cohort of Patients with Inflammatory Bowel Disease

**DOI:** 10.3390/vaccines11101591

**Published:** 2023-10-13

**Authors:** Andrea Costantino, Marco Michelon, Daniele Noviello, Fabio Salvatore Macaluso, Salvo Leone, Nicole Bonaccorso, Claudio Costantino, Maurizio Vecchi, Flavio Caprioli

**Affiliations:** 1Gastroenterology and Endoscopy Unit, Foundation IRCCS Ca’ Granda Ospedale Maggiore Policlinico, 20122 Milan, Italy; maurizio.vecchi@policlinico.mi.it (M.V.); flavio.caprioli@policlinico.mi.it (F.C.); 2Department of Pathophysiology and Transplantation, University of Milan, 20122 Milan, Italy; marco.michelon@unimi.it (M.M.); daniele.noviello@unimi.it (D.N.); 3IBD Unit, Villa Sofia-Cervello Hospital, 90146 Palermo, Italy; fsmacaluso@gmail.com; 4AMICI ETS, 20125 Milan, Italy; salvo.leone@amiciitalia.net; 5Department of Health Promotion Sciences—Maternal and Infant Care—Internal Medicine and Excellence Specialties “G. D’Alessandro”, University of Palermo, 90127 Palermo, Italy; nicole.bonaccorso@unipa.it (N.B.); claudio.costantino01@unipa.it (C.C.)

**Keywords:** inflammatory bowel disease, IBD, Crohn’s disease, ulcerative colitis, vaccine, vaccination, vaccine hesitancy, VZV

## Abstract

Background: The vaccination status of patients with inflammatory bowel disease (IBD) should be investigated before starting any treatment, and patients should eventually be vaccinated against vaccine-preventable diseases (VPDs). Patients with IBD may have suboptimal vaccination rates. The aim of this study was to evaluate the vaccination coverage, attitude towards vaccinations, and determinants among an Italian cohort of patients with IBD. Methods: AMICI, the Italian IBD patients’ association, sent an anonymous web-based questionnaire in February 2021. Previous vaccination status and patients’ attitudes towards vaccinations were recorded. We examined the factors influencing their attitudes using crude and adjusted odds ratios (adjORs) with 95% confidence intervals (CIs). Results: Among the 4039 patients invited, 1252 patients (including 729 women, median age 47.7 [37–58]) completed the questionnaire, with a response rate of 25.3%. Respondents declared being vaccinated against tetanus (74.1%), flu (67.7%; last season), MMR (43.3%), HBV (37.1%), pneumococcus (29.1%), meningitis (20%), HAV (16%), VZV (15.3%), and HPV (7.6%). Complete vaccination history was not remembered by 20.7% of the patients. One thousand one hundred and twelve (88.8%) expressed a positive attitude towards vaccination, 91 (7.3%) were indifferent, and 49 (3.9%) reported being opposed to vaccinations. The belief of a possible return of VPDs with a decline in vaccination coverage rates was the factor most strongly related to a positive attitude towards vaccinations (adjOR 5.67, 95% CI 3.45–9.30, *p*-value < 0.001). Conclusions: A low vaccination rate against some VPDs was found among a national cohort of patients with IBD, despite a generally positive attitude towards vaccinations.

## 1. Introduction

Inflammatory bowel diseases (IBD), namely Crohn’s disease (CD) and ulcerative colitis (UC), are chronic, relapsing inflammatory immune-mediated disorders. Many patients affected by IBD need immunosuppressant therapies, which are known to be associated with a higher risk of contracting opportunistic infectious diseases and of pre-neoplastic or neoplastic lesions such as cervical high-grade dysplasia and cancer [1,2]. Many of these potentially harmful diseases, such as hepatitis B (HBV), flu, chickenpox, herpes zoster virus (HZV), pneumococcal pneumonia, or human papilloma virus (HPV) infection, can be prevented by vaccines [3]. Each drug used in the treatment of IBD should be classified according to the degree of immunosuppression induced in the patient.

According to European Crohn’s and Colitis Organisation (ECCO) guidelines, patients treated with anti-tumour necrosis alpha (antiTNF), corticosteroids, azathioprine, or 6-mercaptopurine and those on combination therapies are at increased infectious risk. Although some of the latest therapies approved for IBD treatments, such as ustekinumab, seem to have a lower degree of immunosuppression compared to anti-TNF drugs, they are not risk-free drugs for infections. For example, patients of all ages treated with tofacitinib are at higher risk of varicella-zoster virus (VZV) infection [4]. Several guidelines suggest investigating patients’ vaccination status before starting any treatment and performing vaccinations against VPDs when required [3,5,6]. 

Compared to the rest of the population, patients affected by IBD are known to be at higher risk of contracting some vaccine-preventable diseases such as flu and pneumonia [7,8].

Nevertheless, despite the increased risk of infections, vaccination rates in IBD patients are known to be suboptimal and may also be lower than vaccination rates in the general population [9,10,11,12,13,14,15,16,17,18].

The National vaccination prevention plan of the Italian Ministry of Health has the following objectives: to achieve and maintain elimination of measles and rubella, to strengthen the prevention of cervical cancer and other HPV-related diseases, to achieve and maintain target vaccination coverage, to promote vaccination interventions in population groups at high risk for pathology, to reduce inequalities and provide actions for population groups that are difficult to reach and/or with low vaccination coverage, to complete the computerization of the regional vaccination registers and implement the national vaccination register, to improve surveillance of vaccine-preventable diseases, to strengthen communication in the field of vaccination, to promote the culture of vaccinations, and to increase knowledge in vaccinology among Healthcare Providers (HCPs) [19].

During the COVID-19 pandemic, the debate around vaccinations in the general population rose to prominence again due to concerns about mass vaccinations. To the best of our knowledge, our study was the first national post-pandemic survey to investigate vaccination coverage against VPDs in IBD patients.

The aim of the study was to investigate vaccination coverage against VPDs, attitudes towards vaccinations, and their possible determinants among a national cohort of IBD patients. Moreover, we aimed to evaluate whether the low vaccination coverage among IBD patients was mainly influenced by vaccination hesitancy or by suboptimal prescription by HCPs.

## 2. Materials and Methods

In February 2021, the Italian IBD patients’ association (Associazione Nazionale per le Malattie Infiammatorie Croniche dell’Intestino, also known as AMICI) distributed an anonymous online questionnaire to their adult members via private and limited mailing lists and social media platforms. The questionnaire was sent in one single mailing without any reminder, and only one post was made on AMICI social media platforms Facebook and Instagram (Meta platforms Inc., Menlo Park, CA, USA). The questionnaire was dedicated only to IBD patients and consisted of an adapted version of a previously validated questionnaire on vaccine hesitancy [20] and was divided into two sections seeking information on: (1) sociodemographic characteristics, lifestyle, and IBD characteristics (gender, age, marital status, educational level, number and age of family members, disease type, adherence to IBD and other therapies/preventive activities, alcohol intake and smoking habit, type of therapy for IBD), and (2) attitude towards vaccinations in general. Patients were asked to self-report their previous vaccinations and their attitudes towards them (with multiple-choice questions). Attitudes towards vaccination were defined: opposed to vaccinations, indifferent, or in favour of vaccinations. The questionnaire was divided into seven sections and is reported in the Appendix A.

### 2.1. Statistical Analysis

In this study, we determined the absolute and relative frequencies for the categorical (qualitative) variables. Means and ranges with 95 CI% were used to describe the quantitative variables. We included in a multivariate backward stepwise logistic regression model all the variables that demonstrated a statistically significant correlation with vaccination attitude in the univariate analysis. All variables with a *p* value ≤ 0.20 were selected in the multivariate model to guarantee a conservative approach. A multiple regression model was used, calculating the crude odds ratio (crude OR) and the adjusted OR (adjOR) with 95% confidence intervals (CIs). The level of significance chosen for the multivariate logistic regression analysis was 0.05 (2-tailed).

Overfitting occurs when too many variables are included in the model and the model appears to fit well with the current data. Overfitting is caused by multiple tests in which some noise variables are entered into the model simply by chance. To overcome this limit, only the variables deemed of interest were entered, and for this reason, the univariate screening of 0.20 was used.

### 2.2. Ethical Statement

The study was approved by the Scientific Advisory Board Ethics Committee of AMICI ETS. All the subjects received an email explaining the rationale of the study and the digital informed consent to participate, and they had to sign the digital informed consent before participating. After they agreed, the subjects were directed via a link to an online structured questionnaire on the SurveyMonkey platform (Momentive Inc., San Mateo, CA, USA) [21].

## 3. Results

The questionnaire was sent to 4720 patients on the AMICI mailing list and had a response rate of 26.5% (1252 patients, including 729 women, median age 47.7, interquartile range 37–58). Fourty-nine percent of participants reported suffering from Crohn’s disease and 48.9% from ulcerative colitis. Completed questionnaires were received from each of the 20 Italian regions. Socio-demographic, lifestyle, and clinical characteristics of the study population are described in Table 1.

Of note, 46.7% of the patients reported being treated with biologic or immunosuppressive drugs.

Patients declared being vaccinated against the following diseases: 74.1% tetanus, 67.7% influenza (during last season), 43.3% (measles, mumps, and rubella) MMR, 37.1% HBV (hepatitis B), 29.1% pneumococcus (pneumococcal conjugated 13-valent vaccine, *PCV13,* or pneumococcal polysaccharide 23-valent vaccine, PPSV23), 20% meningococcal meningitis, 16% HAV (hepatitis A), 15.3% VZV (varicella zoster vaccine), 7.6% HPV (human papillomavirus). Two hundred and fifty-nine (20.7%) did not remember every previous vaccination. Reports of previous vaccination histories are summarised in Table 2.

The participants’ attitudes towards vaccinations in the study population are reported in Table 3.

In summary, among the respondents, 1154 (92.2%) stated they wanted to be vaccinated in the future against VPDs. A previous negative experience with vaccinations, whether personal or referred by relatives, was reported by 163 (13.2%) of the 1238 respondents to this specific question. One thousand one hundred and twelve (88.8%) stated a positive attitude towards vaccination, 91 (7.3%) were indifferent, and 49 (3.9%) reported being opposed to vaccinations. Four hundred and fifty-six (36.4%) stated that the main reason for vaccination adherence was their IBD.

The main determinant associated with a positive attitude towards vaccinations was the belief in the possible return of VPDs with a decline in vaccination coverage rates (adjOR 5.67, 95% CI 3.45–9.30, *p*-value < 0.001). The vaccination adherence motivated by their IBD was at the limit of significance (1.72 (0.99–2.97)) Table 4.

Other factors, such as gender, age, education level, number of family members, marital status, disease type, adherence to the IBD therapy, disease duration, type of therapy (immunosuppressive or not), smoking habit, and use of complementary and alternative medicine, did not influence the attitude towards vaccinations.

## 4. Discussion

Our results show a general positive attitude towards vaccinations, mainly influenced by awareness of the possible return of opportunistic infections with the decline in vaccination rates. Only 3.9% of patients opposed vaccinations. Nevertheless, there is still suboptimal vaccination coverage against VPDs in this national cohort of IBD patients, with almost half of patients (46.7%) taking immunosuppressive or biologic therapies.

Since most patients have a positive attitude towards vaccinations, this suggests a possible role of physicians in under-prescribing vaccinations to this population, possible difficulty in organising vaccinations, and possible low patient awareness.

IBD patients affected by the flu are at higher risk of hospitalisation and developing serious complications.

If we compare the target threshold for flu vaccinations as a comparison model applicable to the target population (65 years old and high-risk individuals of all ages), our population has a reported 67.7%, which is below the 75% threshold with an ideal 95% coverage. In 2021, the coverage against flu was 65.3% for the target population (comparable to our result) and 23.7% for the general population.

IBD patients on immunosuppressive treatment are at higher risk of contracting pneumonia and have an increased mortality rate when hospitalised [7]. For these reasons, annual vaccinations against the flu with inactivated vaccines are recommended for all patients, including those on immunosuppressive therapy. Vaccination against pneumonia is also recommended at the time of IBD diagnosis [5,6]. Nevertheless, the vaccination coverage for flu in these patients is known to be suboptimal [8,11]. Other potentially harmful VPDs include the *Neisseria meningitidis* infection, which can cause meningitis with a high risk of complications. Anti-meningococcal vaccination can be safely administered to IBD patients independently of the therapies they are taking [5,6]. Before starting immunosuppressive treatment, the immunisation status of IBD patients for other diseases, including HPV, HBV, and VZV, should also be checked [3,5,6]. HPV can cause anogenital and cervical cancer; therefore, both men and women should be encouraged to get vaccinated before starting any treatment for IBD. Moreover, all women under immunosuppressants or steroid therapy should be encouraged to participate in cervical cancer screening at a higher frequency than the rest of the population because of the reported increased risk of cervical cancer precursor lesions [12,13,14]. Since 2007, HPV vaccination has been offered actively and free of charge to girls >12 years old; every Italian region is allowed to include additional age cohorts as target groups in the HPV vaccination programme; since 2014, many regions have extended the active vaccination programme to boys aged >12 years old.

The national plan for the elimination of measles, mumps, and rubella provides for the vaccination of women of childbearing age if they are not naturally immunised in the immunological test.

HBV antibody titers should be tested before starting any patient on immunosuppressive treatment due to the risk of HBV reactivation, which can result in hepatitis and hepatic failure. Patients with an antibody titer <10 mUI/mL, particularly, should be vaccinated or revaccinated, according to national or regional guidelines [3].

It is well known that IBD patients, particularly those on treatment with immunosuppressives, biological drugs, or small molecules, have a greater risk of severe primary VZV infection and of herpes zoster (HZ). The former can be life-threatening in immunocompromised patients, while the latter most frequently affects patients aged over fifty and those on combined immunosuppressive treatment. For this reason, vaccination against VZV is recommended for IBD patients receiving immunosuppressive therapy [15,16,17]. Recombinant herpes zoster vaccine (RZV) is the preferred vaccine for patients with IBD disease, given its efficacy and safety. If RZV is not available, a live zoster vaccine [ZVL] is recommended in immunocompetent patients with IBD aged ≥50 years [3].

This study has some limitations that must be addressed. First, people who filled out the questionnaire could have had a greater willingness to be vaccinated than those who did not answer, representing a possible response bias. A similar limitation could be the possible selection bias arising from the delivery of the questionnaire through the mailing list of AMICI ETS. Those patients may be more concerned about their disease or have a more proactive attitude. Another possible limitation of the study is that the median age of the respondents was 47.7. There is a possible higher inclination towards vaccination in middle-aged patients due to heightened fears of complications related to infectious diseases. A major limitation of our survey is the use of a self-reported questionnaire instead of vaccination cards or official records. This may represent a recall bias; of note, 20.7% of the respondents could not remember every previous vaccination, despite all being members of a patients’ association. Despite this, a questionnaire still represents the most efficient way to investigate not only the vaccination status but also a great deal of data and parameters regarding vaccination hesitancy in a national cohort in such a short period of time. Furthermore, studies that compared the accuracy of self-reported vaccination status with official records showed comparable results [22,23]. In 2021, Smith et al. demonstrated that self-reporting was an effective way to determine flu immunisation status, which provided useful information prior to administering pneumococcal vaccines to patients with IBD [22]. Another study conducted on smallpox vaccines indicated substantial and acceptable agreement between participants self-reporting of vaccination status and electronic documentation [23].

Despite these limitations, our survey has many strengths.

First, every patient filled out the questionnaire deliberately and without remuneration. At the time of writing, this study represented the first post-pandemic national survey investigating vaccine status and attitude towards vaccination in a national IBD cohort. The questionnaire was sent through the mailing list of the major national patients’ association, so it gives a realistic picture of the vaccination status among the Italian IBD population. The questionnaire was an adapted version of a previously validated questionnaire on vaccine hesitancy [20]. It investigated vaccination attitudes through several sections and had a low response rate but was comparable to other web-based surveys (~25%). The greatest strength of our survey compared to previous studies investigating vaccine status in IBD patients is the high number of respondents (*n* = 1252). This represents one of the biggest IBD populations investigated, both for their vaccination coverage and hesitancy. The high response rate is likely due to the concurrent investigation of hesitancy and attitudes towards COVID-19 vaccines when the vaccination campaign started in Italy.

In the future, vaccination status among the IBD population could be examined through national official records or in multicentre cohorts of patients. Such studies should carefully consider issues around patients’ privacy.

In our survey, however, 88.8% of the participants stated a positive attitude towards vaccination, while only 3.9% of the respondents expressed a negative attitude towards vaccines. This is a very encouraging result, as other studies investigating vaccine hesitancy among patients affected by chronic illnesses showed more negative attitudes [24].

Many IBD patients may be hesitant towards vaccines because of concerns about the balance between the safety and benefits of vaccination [25]. Many studies have demonstrated that vaccine hesitancy is a common phenomenon globally, with variability in the rationale behind refusal of vaccine acceptance, including perceived risks and benefits, religious beliefs, and a lack of knowledge and awareness [26,27,28].

According to the Strategic Advisory Group of Experts on Immunization (SAGE), vaccine hesitancy is the term used to describe: “delay in acceptance or refusal of vaccination despite availability of vaccination services”. Attitude towards vaccination is affected by many factors, including complacency, convenience, and confidence. [26] Other factors associated with vaccine hesitancy include public health policies, social factors, previous experience, educational and income levels, and the messages spread by the media [28].

The COVID-19 pandemic brought back to public debate the discussion around vaccine hesitancy because of some mass media misinformation and emphasis on the hypothetical side effects of vaccines, including long-term side effects, the toxicity of adjuvants and preservatives, and the weakening of the immune system [27].

In our survey, however, 88.8% of the participants stated a positive attitude towards vaccination, while only 3.9% of the respondents expressed a negative attitude towards vaccines. This is a very encouraging result, as other studies investigating vaccine hesitancy among patients affected by chronic illnesses showed more negative attitudes [24]. Comparable results were found between patients who use chronic immunosuppressive treatments and those who underwent liver transplantation [29].

Patients in our study stated that their positive attitude towards vaccinations was mainly influenced by their awareness of the potentially harmful opportunistic infections that could spread again with the decline of vaccination coverage rates. Unexpectedly, the attitude towards vaccination was not influenced by the immunosuppressive treatments.

Unfortunately, despite a general positive attitude towards vaccinations, our study showed that the vaccination coverage among IBD patients keeps on being suboptimal, as previously shown in many pre-pandemic studies [9,10,11,12,13,14,15,16,17,18]. There is often poor awareness of the importance of vaccines for IBD patients by patients themselves as well as by gastroenterologists and general practitioners [30,31].

Gastroenterologists are the primary HCPs for IBD patients, playing a key role in ensuring adequate disease management. Unfortunately, their knowledge on the correct use of vaccines is often insufficient [30,31,32], and they do not always provide adequate patient counselling [10], as has emerged from specific surveys.

Since the general lack of attention towards the importance of vaccinations could be due both to general practitioners and IBD specialists, the best way in which physicians can be helped is by the provision of checklists expressly created to investigate patients’ vaccination coverage and vaccinations to be prescribed. Several guidelines suggest that patients’ vaccination status should be checked by physicians at the time of diagnosis. This is particularly the case for gastroenterologists, who play a primary and pivotal role in the treatment of IBD patients. A vaccination plan should be defined before starting any immunosuppressive treatment. It is essential to keep on promoting and updating guidelines that provide specific indications on how to actively advocate for vaccination, particularly for those IBD patients who need immunosuppressive therapies. As some gastroenterologists think that the planning and administering of vaccines should be performed by general practitioners, a good strategy to increase vaccination coverage among IBD patients could be improving the communication between these categories of health care providers.

Other strategies include the involvement of patients’ associations in spreading a culture of vaccination, the implementation of awareness campaigns aimed at adolescents, young adults, and adults with the support of digital instruments, and the organisation of specific vaccination events. Web-based surveys such as this one could represent a good awareness instrument. Other web-based instruments such as telemedicine could play an important role, in particular during pandemics.

For those patients with a greater probability of being hesitant against vaccines (e.g., complementary and alternative medicine users, patients with low education levels), the best way to change their minds may be by optimising patient-doctor communication. Finally, education and correct information still represent the best ways to improve vaccination coverage among IBD patients. Even when an adequate level of awareness is present, messages and warnings from healthcare providers seem to be necessary.

## 5. Conclusions

The public debate around vaccinations is a trending topic. To the best of our knowledge, this study represents the first national post-pandemic survey to investigate coverage and attitudes towards general vaccinations among IBD patients.

Our study demonstrated that, despite a general positive attitude towards vaccinations mainly influenced by awareness of the possible return of opportunistic infections with and decline in vaccination rates, there is still suboptimal vaccination coverage against VPDs in IBD patients.

This might suggest a possible role of physicians in under-prescribing vaccinations to this population, since most patients have a positive attitude. A minority of hesitant patients (3.9%) still remain unconvinced, and this may be overcome primarily by enhanced patient-doctor communication.

The results of this survey could be a starting point for developing specific vaccination campaigns to increase vaccination rates against VPDs in IBD patients.

## Figures and Tables

**Table 1 vaccines-11-01591-t001:** Socio-demographic, lifestyle, and clinical characteristics of the study population with inflammatory bowel diseases (IBD) (*n* = 1252).

Characteristics	Number (*n*)	Percentage (CIs 95%)
Gender	1252	
Male	523	41.8 (39–44.6)
Female	729	58.2 (55.4–61)
Age (years), mean (range)	47.7 (37–58)	
Marital status		
Married/cohabitant/second marriage	854	68.2 (65.4–71.6)
Single/divorced/widowed	398	31.8 (28.4–34.6)
Educational level		
Undergraduate	778	62.1 (59.4–64.8)
Graduate	474	37.9 (35.2–40.6)
Number of family members		
<4	1205	96.2 (94.8–97.9)
>4	47	3.8 (2.1–5.2)
Children under 10 years of age		
No	1065	85.1 (81.4–89)
Yes	187	14.9 (11–18.6)
Disease type		
Crohn’s Disease	614	49 (46.2–51.8)
Ulcerative colitis	612	48.9 (46.1–51.7)
Indeterminate colitis	26	2.2 (1.5–3.2)
Adherence to therapy recommended for IBD		
No	16	1.3 (0.8–2.1)
Yes/Most of the time	1236	98.7 (97.9–99.2)
Therapy		
None/mesalamine	668	53.3 (50.5–56.1)
Biologic or immunosuppressive drug	584	46.7 (43.9–49.5)
Disease duration		
<5 Years	161	12.9 (11.1–14.3)
>5 Years	1091	87.1 (85.7–88.9)
Working as healthcare professionals (HCPs)		
No	1100	87.9 (85.9–89.6)
Yes	152	12.1 (10.4–14.1)
Adherence to other preventive activities (e.g., oncological screening)		
No	355	28.4 (25.9–31)
Yes	897	71.6 (69–74.1)
Alcohol intake		
No	759	60.6 (57.8–63.3)
Yes often/minimal consumption	493	39.4 (36.7–42.2)
Self-reported active lifestyle		
No	670	53.5 (50.7–56.3)
Yes	582	46.5 (43.7–49.3)
Smoking habit		
Non-smoker	899	71.8 (69.2–74.3)
Smoker/former smoker	353	28.2 (25.7–30.8)
Use of complementary and alternative medicines (CAMs)		
No	1093	87.3 (85.3–89.1)
Yes	159	12.7 (10.9–14.7)

**Table 2 vaccines-11-01591-t002:** Reported previous vaccinations among the study population affected (*n* = 1252).

Previous Vaccines	Number (*n*)	Percentage (CIs 95%)
Tetanus	928	74.1 (71.5–77.3)
HBV (hepatitis B)	464	37.1 (35.2–39)
HAV (hepatitis A)	200	16 (13.9–17.7)
MMR (measles, mumps, rubella)	542	43.3 (41.1–45.4)
Influenza	848	67.7 (64.9–69.1)
Pneumococcus (PCV13 and/or PPSV23)	364	29.1 (27.1–30.9)
HPV (human papillomavirus)	95	7.6 (6.3–8.5)
Meningococcal meningitis	250	20 (18.4–21.7)
VZV (varicella zoster vaccine)	192	15.3 (12.9–17)
Patients were vaccinated for some of the reported VPDs	255	20.4 (18.7–22.1)
Patients were never vaccinated against the reported VPDs	13	1 (0.2–2.3)
Patients who did not remember previous vaccinations	259	20.7 (18.4–23.5)

Note: HBV, Hepatitis B; HAV, Hepatitis A; MMR, measles, mumps, and rubella; *PCV13*, pneumococcal conjugated 13-valent vaccine; PPSV23, pneumococcal polysaccharide 23-valent vaccine; HPV, human papillomavirus; varicella-zoster virus, VZV; VPDs, vaccine-preventable diseases.

**Table 3 vaccines-11-01591-t003:** Attitudes towards vaccinations in the study population (*n* = 1252).

Attitudes towards Vaccinations	Number (*n*)	Percentage (Cis 95%)
Opposed to vaccinations	49	3.9 (2.9–5.2)
Indifferent to vaccinations	91	7.3 (5.9–8.9)
In favour of vaccinations	1112	88.8 (86.9–90.5)
Willingness to be vaccinated in the future (against COVID-19 and other diseases)		
No	98	7.8 (6.4–9.5)
Yes	1154	92.2 (90.5–93.6)
Willingness to vaccinate your children in the future	(938)	
No	24	2.6 (1.7–3.1
Yes, totally	752	80.2 (78.6–82.7)
Yes, partially	162	17.2 (15.6–18.3)
Possible return of VPDs with a decline in vaccination coverage rates		
No	160	12.8 (11.4–14.5
Yes	1092	87.2 (85.5–88.6)
Previous negative experience (personal/family members/relatives reported/referred) with vaccinations	(1238)	
No	1075	86.8 (84.9–88.3)
Yes	163	13.2 (11.7–15.1)
Best strategy to prevent VPDs		
Vaccination	624	49.8 (47.4–51.1)
Other (diet, physical activity, homeopathy, etc.)	60	4.8 (4.3–6.1)
Vaccination and other strategies	568	45.4 (44.6–46.5)
Main reason for vaccination adherence due to their IBD		
No	796	63.6 (61.8–65.2)
Yes	456	36.4 (34.8–38.2)
Higher confidence in HCPs in comparison with mass media on vaccine information	(1227)	
No	51	4.2 (3.5–5.4)
Yes	1176	95.8 (94.6–96.5)

Note: HCPs, healthcare professionals, VPDs, vaccine-preventable diseases.

**Table 4 vaccines-11-01591-t004:** Crude OR and adjOR of factors associated with trust and positive attitude regarding vaccinations among patients with IBD enrolled in the study (*n* = 1252).

	Crude OR	CI 95%	*p*-Value	adjOR	CI 95%	*p*-Value
Gender						
*Male*	*ref*		0.92			
*Female*	0.98	(0.69–1.40)				
Age in years (continuous variable)	0.99	(0.98–1.01)	0.94			
Marital Status						
*single/divorced/widowed*	*ref*		0.42			
*married or cohabitant*	0.95	(0.82–1.08)				
Children under 10 years of age						
*No*	*ref*		0.69			
*Yes*	1.04	(0.87–1.22)				
Degree						
*Under graduation*	*ref*		<0.05	*ref*		0.29
*Graduation*	1.59	(1.01–2.53)		1.38	(0.75–2.52)	
Working as a healthcare professionals						
*No*	*ref*		0.78			
*Yes*	1.08	(0.63–1.87)				
Adherence to therapy recommended for IBD						
*No*	*ref*		0.35			
*Yes*	2.58	(0.48–9.35)				
Smoking habit						
*No*	*ref*		0.87			
*Yes*	0.75	(0.51–1.09)				
Physical activity						
*No*	*ref*		0.60			
*Yes*	0.91	(0.64–1.29)				
Alcohol use						
*No*	*ref*		0.21			
*Yes*	1.26	(0.88–1.81)				
Use of homeopathic products; belief in alternative medicine						
*No*	*ref*		<0.01	*ref*		0.27
*Yes*	0.48	(0.31–0.75)		0.89	(0.67–1.18)	
Adherence to other preventive activities (e.g., oncological screening)						
*No*	*ref*		<0.05	*ref*		0.1
*Yes*	1.58	(1.09–2.28)		1.43	(0.89–2.31)	
Possible return of vaccine-preventable diseases (VPDs) with a decline in vaccination coverage rates						
*No*	*ref*		<0.001	*ref*		<0.001
*Yes*	11.3	(7.64–16.9)		5.67	(3.45–9.30)	
Past negative experience (also reported/referred) with vaccination						
*No*	*ref*		<0.001	*ref*		0.16
*Yes*	0.26	(0.17–0.39)		0.66	(0.36–1.18)	
Main reason for vaccination adherence is due to IBD						
*No*	*ref*		<0.001	*ref*		0.053
*Yes*	3.43	(2.14–5.49)		1.72	(0.99–2.97)	
Higher confidence in HCPs in comparison with mass media on vaccine information						
*No*	*ref*		<0.001	*ref*		0.07
*Yes*	3.30	(1.73–6.27)		2.33	(0.93–5.81)	
Immunosuppressive therapies						
*No*	*ref*		<0.029	*ref*		0.179
*Yes*	2.18	(1.07–3.79)		1.35	(0.74–4.75)	

Note: HCPs, healthcare professionals, VPDs, vaccine-preventable diseases.

## Data Availability

The datasets generated and analyzed during the current study are not publicly available but are available from the corresponding author on reasonable request.

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
