# Peer review of "Attitudes towards Vaccinations in a National Italian Cohort of Patients with Inflammatory Bowel Disease"

_vaccines, 2023, doi:10.3390/vaccines11101591_

Round 1

Reviewer 1 Report

This manuscript is good; however, it requires extensive grammar editing and English too. Please check the complete manuscript carefully in the revisions.

The discussion section should be continuous hence remove the subsections.

This manuscript is good; however, it requires extensive grammar editing and English too. Please check the complete manuscript carefully in the revisions.

The discussion section should be continuous hence remove the subsections.

Author Response

Dear reviewer,
Thank You very much for your positive comment on the quality of the manuscript.
The Manuscript has been reviewed by a certified native English-speaking copy editor.

We removed the subsections in order to have a continous discussion, as You suggested.

Thank You.

Best regards.

Reviewer 2 Report

The authors describe the results of vaccine’s coverage in an Italian’s cohort of patients suffering from IBD. The text is well written and of importance in the context of vaccine’s hesitancy but the methods and presentation of the results need to be improved and I have some questions and comments that should be answered before the paper can be accepted for publication.

Abstract:

          You should include the CI for the prevalence of vaccination

Introduction:

-          Well written and concise, to the points. I would add some information about the Italian healthcare system to better put this study in context (access to care, reimbursement, vaccination, prevention, etc).

Methods:

-          List of the variables collected should be provided

-          statistical analysis: Did the authors check the specification of the multivariate model (overfitting?)

-          The authors stated they used a backward stepwise regression but it seems that they only add the variables with a p-value < 0.2 in the univariate analysis and did not stepwise regression (as most of the variables in the multivariate model are not significant).

Results:

-          In all tables: remove the three last columns

-          Please compute the 95%CI for all the prevalence, Table 3 and 4 present the same information, Table 4 is more informative.

-          Authors should perform subgroup analysis to see if the number are different in the group opposed/indifferent Vs. in favour of the vaccination. Why are these data not in the model?

Discussion:

-          Authors only present the results in their cohort of patients, making it difficult to compare the effect of IBD. It would be interesting to have the number of vaccine coverage in the healthy population so determine whether or not there is a difference.

-          Authors should also discuss targeted threshold for the different vaccination to determine how far (or close) we are from this ideal situation.

-          Limitation: please name de type of bias (selection bias, recall bias) and weaknesses self-reported online questionnaire, authors mentioned a high response rate – according to me it’s quite low (25%).

Well written

Author Response

Dear reviewer, 

Thank You very  much for the time spent and effort spent to review our Manuscript. 
We accepted Your very precise comments with great interest and changed that in the manuscript (underlined). 

Abstract: 
we added CI of prevalence

Introduction: 
thanks for your compliments. As You suggested, we added some information regarding the Italian healthcare system as regards access to vaccinations and vaccination campaigns)

Methods: 

  • A list of the sociodemographic variables collected was added to the main text
  • We thank you for your very precise comment. Only the variables deemed of interest were entered and for this reason the univariate screening of 0.20 was used to contrast overfitting 
  • We thank you for your very precise comment. We apologize, it is a simple multiple regression model having first eliminated the not significant variables and with p.-value greater than 0.20. 

Results:

  • We removed the last three columns that were accidentally added by copy editor in all tables
  • We compute the 95%CI for all the prevalence. Table 4 is thought to evaluate determinants of vaccine hesitancy
  • Thank you for the precise comment, analysis was performed and did not resulted significant (probably due to the number of 49)

Discussion:

  • Thank you very much for your precise comment, we compared even if different population with the flu vaccination data of 2021 both for target population and general population
  • We also added data regarding the threshold for flu vaccination for target and general population in Italy in order to have a comparison
  • we named the type of bias and evaluated the 25% response bias as low. 

We thank You once again very much for Your very precise revisions. 

Best regards. 

Reviewer 3 Report

I've been invited to review the following paper "Attitudes towards Vaccinations in a National Italian Cohort of Patients with Inflammatory Bowel Disease". In this cross-sectional web-based study, Costantino A et al. inquiry the self-declared vaccination status, and the potential barriers/facilitators towards vaccinations in a sample of 1252 subjects with previous diagnosis of IBD. In the end, only the fear for the return of otherwise vanished VPD has been characterized as a notable effector (a positive one). The vaccination rates were often far from optimal, and signs of vaccine hesitancy were substantial. 

Unfortunately, from my point of view, several implementation are suggested before the eventual acceptance of the paper. More precisely:

1) "Between February 10 and February 19, 2021, the Italian IBD patients’ association (Asociazione Nazionale per le Malattie Infiammatorie Croniche dell’Intestino, also known as AMICI) distributed an anonymous online questionnaire to their adult members via mailing lists and social media platforms" but in the next section: "The questionnaire was sent in one single mailing without any reminder, and only one post was made on AMICI social media platforms Facebook and Instagram". The second statement is quite more problematic, as it is unclear whether non-IBD patients could have been invited to participate into this study. Were the social pages of AMICI private or with an access limited by invitation in order to restrain participation to patients (and therefore to suitable participants)? Otherwise, please discuss this shortcoming.

2) According to results section, "The questionnaire was sent to 4720 patients and had a response rate of 26.5%". Participation rate was calculated on the total of members of AMICI? on the total of member of AMICI + members of social pages? on the total of members on mailing list? Please explain.

3) A flow chart reporting the progressive definition of the sample would improve the overall quality of the paper.

4) All tables have retained the column from the original format of the MDPI template, please edit and remove columns as needed.

5) When discussing data on HPV and MMR, it should be of some interest report to the international readers the different gender-based strategies for vaccination in Italy across the last three decades - that could have been among the causes for some of these results.

6) Table 2 reports the term "meningitis"; I guess Authors are meaning Meningococcal meningitis. Please be aware that many adults often associate the term "meningitis vaccine" with "Hemophilus influenzae", while a substantial share of meningitides are not actually meningococcal-caused but pneumococcal-associated. In other term, please explain more accurately what questionnaire hinted in order to report more precisely.

7) the statement from row 171 (All these problems may be solved by specific vaccination campaigns among both HCPs and IBD patients, aimed at increasing vaccination rates against VPDs) is acceptable and otherwise shared by the present reviewer, but the data reported by this study do not support this conclusion so directly. I suggest to remove this sentence of the main text, maintaining the similar content in subsequent section of the main text, as coming after a more extensive discussion.

The main text is of acceptable quality.

Author Response

Dear reviewer, 

Thank You very  much for the time spent and effort spent to review our manuscript. 
We accepted Your very precise comments with great interest and changed that in the manuscript (underlined). 

1) We thank You for the precise comment. We discussed better in the methods. Only IBD patients participated in the questionnaire. AMICI mailing list was only for IBD patients. 

2) 4720 is the number of email address, we precised this in the methods, thanks for the precise comment

3) the visual abstract of this manusctipthas the flow chart of the respondents. We hope this is acceptable to better understand the main message of the article

4) We deleted the 3 columns in the tables (error of the copy editor) not present in the submitted manuscript. 

5) Thank You very much for your precise comment. We added a paragraph regarding the different gender-based strategies for vaccination in Italy across the last three decades

6) We were more precise and indicated the meningococcal meningitidis vaccine as you correctly underlined

7) We do agree that sentence could be removed frome that position and we did as suggested. Thank You very much. 

Thank You for the comment on the English quality. 

Best regards. 

Round 2

Reviewer 1 Report

All the necessary changes have been made; hence the revised manuscript may be accepted in its present form.